# Nearest Neighbor Machine Translation is Meta-Optimizer on Output Projection Layer

**Ruize Gao**[1*]   **Zhirui Zhang**[2†]   **Yichao Du**[3]   **Lemao Liu**[2]   **Rui Wang**[1†]

[1]Shanghai Jiao Tong University    [2]Tencent AI Lab
[3]University of Science and Technology of China
[1]ruizgaonlp@gmail.com, wangrui12@sjtu.edu.cn
[2]zrustc11@gmail.com, redmondliu@tencent.com
[3]duyichao@mail.ustc.edu.cn

## Abstract

Nearest Neighbor Machine Translation ($k$NN-MT) has achieved great success in domain adaptation tasks by integrating pre-trained Neural Machine Translation (NMT) models with domain-specific token-level retrieval. However, the reasons underlying its success have not been thoroughly investigated. In this paper, we comprehensively analyze $k$NN-MT through theoretical and empirical studies. Initially, we provide new insights into the working mechanism of $k$NN-MT as an efficient technique to implicitly execute gradient descent on the output projection layer of NMT, indicating that it is a specific case of model fine-tuning. Subsequently, we conduct multi-domain experiments and word-level analysis to examine the differences in performance between $k$NN-MT and entire-model fine-tuning. Our findings suggest that: (*i*) Incorporating $k$NN-MT with adapters yields comparable translation performance to fine-tuning on in-domain test sets, while achieving better performance on out-of-domain test sets; (*ii*) Fine-tuning significantly outperforms $k$NN-MT on the recall of in-domain low-frequency words, but this gap could be bridged by optimizing the context representations with additional adapter layers.[1]

## 1   Introduction

In recent years, Nearest Neighbor Machine Translation ($k$NN-MT) and its variants (Khandelwal et al., 2021; Zheng et al., 2021a,b; Jiang et al., 2021; Wang et al., 2022a) have provided a new paradigm and achieved strong performance for fast domain adaptation through retrieval pipelines. Unlike model fine-tuning, which requires additional parameter updates or introduces external adapter layers, $k$NN-MT combines traditional Neural Machine Translation (NMT) models (Bahdanau et al., 2015; Vaswani et al., 2017; Hassan et al., 2018) with a token-level $k$-nearest-neighbour retrieval mechanism. This allows for direct access to domain-specific datastores, improving translation accuracy without the need for supervised fine-tuning. Although $k$NN-MT has achieved great success in domain adaptation tasks, its working mechanism is still an open problem that has not been thoroughly investigated.

In this paper, we propose a novel perspective to understand $k$NN-MT by describing it as a special case of fine-tuning, specifically a process of meta-optimization on the Output Projection Layer (OPL) of NMT, and establish connections between $k$NN-MT and model fine-tuning (Section 3). Our novel perspective on $k$NN-MT posits that (*i*) the working mechanism of $k$NN-MT is to implicitly execute gradient descent on OPL, producing meta-gradients via forward computation based on $k$-nearest-neighbors, and (*ii*) explicit fine-tuning on OPL shares a similar gradient format with the meta-gradients obtained by $k$NN-MT, according to the derivation of back-propagation. As illustrated in Figure 1, $k$NN-MT and explicit OPL fine-tuning share a dual view of gradient descent-based optimization. The key difference between them lies in the method for computing gradients: $k$NN-MT produces meta-gradients through forward computation and interpolation, while fine-tuning method computes gradients of OPL via back-propagation. Hence, it is reasonable to understand $k$NN-MT as an implicit form of model fine-tuning.

To provide empirical evidence for our understanding, we carry out experiments based on multi-domain datasets (Section 4.1). Specifically, we compare the model predictions of $k$NN-MT and explicit OPL fine-tuning on five domain adaptation tasks. As expected, the predictions of $k$NN-MT is highly similar to that of explicit OPL fine-tuning. These findings support our understanding that $k$NN-MT performs implicit OPL fine-tuning.

---

*This work was done when Ruize Gao was interning at Tencent AI Lab.

†Corresponding authors.

[1]Our code is open-sourced at https://github.com/RuizGao/knnmt-meta-optimizer.

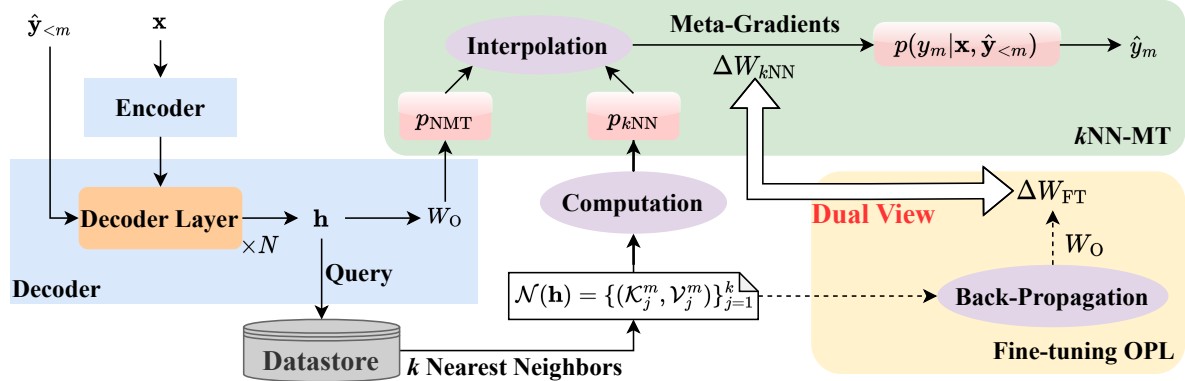

Figure 1: $k$NN-MT implicitly executes gradient descent on the Output Projection Layer (OPL) of NMT and produces meta-gradients via forward computation based on $k$-nearest-neighbors. The meta-optimization process of $k$NN-MT shares a dual view with explicit OPL fine-tuning that updates the parameters of OPL with back-propagated gradients.

Next, we conduct comprehensive multi-domain experiments and word-level analysis to examine the differences in translation performance between $k$NN-MT and other popular fine-tuning methods, such as entire-model fine-tuning and adapter-based fine-tuning (Section 4.2 and 4.3). Our empirical results suggest that: (*i*) Introducing $k$NN-MT on top of adapter-based fine-tuning obtains comparable translation performance to entire-model fine-tuning on in-domain test sets, while achieving better performance on out-of-domain test sets. (*ii*) The entire-model fine-tuning significantly outperforms $k$NN-MT in terms of the recall of in-domain low-frequency words, but this difference can be mitigated by optimizing the context representations with lightweight adapter layers.

## 2 Background

### 2.1 Neural Machine Translation

NMT employs an encoder-decoder model with neural networks that are parameterized by $f_\theta$ to establish the mapping between the source sentence $\mathbf{x}$ and its corresponding target sentence $\mathbf{y}$. For the decoding stage, at time step $m$, NMT utilizes the context representation $\mathbf{h} \in \mathbb{R}^{d_{\text{in}}}$, which is generated from the source sentence $\mathbf{x}$ and the current target context $\hat{\mathbf{y}}_{<m}$, to predict the next-token probability:

$$\mathbf{h} = f_\theta(\mathbf{x}, \hat{\mathbf{y}}_{<m}),$$
$$p_{\text{NMT}}(y_m|\mathbf{x}, \hat{\mathbf{y}}_{<m}) = \text{softmax}(W_{\text{O}}\mathbf{h}), \quad (1)$$

where $W_{\text{O}} \in \mathbb{R}^{|\mathcal{Y}| \times d_{\text{in}}}$ represents the parameter matrix of OPL in the NMT model and $|\mathcal{Y}|$ is the vocabulary size.

### 2.2 Nearest Neighbor Machine Translation

Khandelwal et al. (2021) propose $k$NN-MT that enhances pre-trained NMT models on the general domain by incorporating a translation memory retriever. It enables the models to leverage external in-domain knowledge and improve the quality of in-domain translations. This approach is generally formulated in two processes: datastore construction and inference with $k$NN retrieval.

The datastore is a translation memory that converts bilingual sentence pairs into a set of key-value pairs. For a given target domain bilingual corpus $\{(\mathbf{x}, \mathbf{y})\}$, the context representation $f_\theta(\mathbf{x}, \mathbf{y}_{<m})$ generated by the pre-trained NMT model at each timestep $m$ is used as the key, and the $m$-th target token $y_m$ is treated as the corresponding value, resulting in a key-value pair. The entire corpus contributes to the datastore $\mathcal{D}$, which is comprised of all key-value pairs:

$$\mathcal{D} = \bigcup_{(\mathbf{x}, \mathbf{y})} \{(f_\theta(\mathbf{x}, \mathbf{y}_{<m}), y_m), \forall y_m \in \mathbf{y}\}. \quad (2)$$

During inference, the model utilizes the current context representation $\mathbf{h} = f_\theta(\mathbf{x}, \hat{\mathbf{y}}_{<m})$ at the $m$-th decoding step to produce a probability distribution over a restricted vocabulary obtained through a nearest-neighbor approach:

$$p_{k\text{NN}}(y_m|\mathbf{x}, \hat{\mathbf{y}}_{<m}) \propto$$
$$\sum_{(\mathcal{K}_j^m, \mathcal{V}_j^m) \in \mathcal{N}(\mathbf{h})} \mathbb{1}_{y_m = \mathcal{V}_j^m} \cdot \exp(\frac{d(\mathcal{K}_j^m, \mathbf{h})}{T}),$$
$$(3)$$

where $T$ denotes the temperature to control the sharpness of the softmax function and $\mathcal{N}(\mathbf{h}) = \{(\mathcal{K}_j^m, \mathcal{V}_j^m)\}_{j=1}^k$ is the set of $k$ nearest-neighbors

retrieved from $\mathcal{D}$ using a pre-defined distance function $d(.,.)$. In practice, we can use either the dot-product function or negative $l_2$ distance to implement $d(.,.)$. Xu et al. (2023) have demonstrated that the performance of these two functions is almost identical, so we adopt the dot-product function for theoretical analysis in this paper. Finally, $k$NN-MT interpolates the vanilla NMT prediction $p_{\text{NMT}}$ with the $k$NN prediction $p_{k\text{NN}}$ to obtain the final next-token probability:

$$
\begin{aligned}
p(y_m|\mathbf{x}, \hat{\mathbf{y}}_{<m}) = {}& \lambda \cdot p_{k\text{NN}}(y_m|\mathbf{x}, \hat{\mathbf{y}}_{<m}) \\
& + (1-\lambda) \cdot p_{\text{NMT}}(y_m|\mathbf{x}, \hat{\mathbf{y}}_{<m}),
\end{aligned}
\tag{4}
$$

where $\lambda$ is a tuned interpolation coefficient. In addition, this prediction way could also be substituted with other $k$NN variants (Zheng et al., 2021a; Wang et al., 2022a; Dai et al., 2023b) to achieve better model performance or inference speed.

## 2.3 Dual Form Between Gradient Descent Based Optimization and Attention

Irie et al. (2022) present that linear layers optimized by gradient descent have a dual form of linear attention, which motivates us to view $k$NN-MT as meta-optimizers. Concretely, a linear layer optimized via gradient descent can be formulated as:

$$
\mathcal{F}(\mathbf{q}) = (W_0 + \Delta W)\mathbf{q},
\tag{5}
$$

where $\mathbf{q} \in \mathbb{R}^{d_{\text{in}}}$ is the input representation, and $W_0, \Delta W \in \mathbb{R}^{d_{\text{out}} \times d_{\text{in}}}$ are the initialized parameter matrix and the updated matrix, respectively. In the back-propagation algorithm, $\Delta W$ is computed by accumulating $n$ training inputs to this layer $\mathbf{Q} = (\mathbf{q}_1, ..., \mathbf{q}_n) \in \mathbb{R}^{d_{\text{in}} \times n}$ and corresponding (back-propagation) error signals $\mathbf{E} = (\mathbf{e}_1, ..., \mathbf{e}_n) \in \mathbb{R}^{d_{\text{out}} \times n}$ obtained by gradient descent:

$$
\Delta W = \sum_{i=1}^{n} \mathbf{e}_i \otimes \mathbf{q}_i = \mathbf{E}\mathbf{Q}^\top.
\tag{6}
$$

The dual form of a linear layer trained by gradient descent is a key-value memory with attention storing the entire training experience:

$$
\begin{aligned}
\mathcal{F}(\mathbf{q}) &= (W_0 + \Delta W)\mathbf{q} \\
&= W_0\mathbf{q} + \mathbf{E}\mathbf{Q}^\top \mathbf{q} \\
&= W_0\mathbf{q} + \text{LinearAttn}(\mathbf{Q}, \mathbf{E}, \mathbf{q}),
\end{aligned}
\tag{7}
$$

where $\text{LinearAttn}(\mathbf{K}, \mathbf{V}, \mathbf{q})$ denotes the linear attention operation, and we regard the training inputs $\mathbf{Q}$ as keys, the error signals $\mathbf{E}$ as values, and the

current input $\mathbf{q}$ as the query. Instead of using the regular softmax-normalized dot product attention, which is $\text{Attention}(\mathbf{K}, \mathbf{V}, \mathbf{q}) = \mathbf{V}\text{softmax}(\mathbf{K}^\top \mathbf{q})$, we investigate the working mechanism of $k$NN-MT under a relaxed linear attention form, following the approach of Irie et al. (2022).

## 3 $k$NN-MT Performs Implicit Gradient Descent on Output Projection Layer

In this section, we first demonstrate that probability distribution in $k$NN-MT, including $p_{k\text{NN}}$ and $p_{\text{NMT}}$, is equivalent to Transformer attention. On top of that, we argue that $k$NN-MT implicitly performs gradient descent on OPL, producing meta-gradients via forward computation and interpolation based on $k$-nearest-neighbors. Next, we draw comparisons between $k$NN-MT and explicit OPL fine-tuning, establishing connections between these two forms.

### 3.1 Output Distributions are Attentions

Let $\mathbf{h} = f_\theta(\mathbf{x}, \hat{\mathbf{y}}_{<m})$ be the context representation at each timestep $m$, and we obtain the nearest neighbors set $\mathcal{N}(\mathbf{h}) = \{(\mathcal{K}_j^m, \mathcal{V}_j^m)\}_{j=1}^k$ from the datastore $\mathcal{D}$. Let $\boldsymbol{\mathcal{K}}_m = [\mathcal{K}_1^m, \mathcal{K}_2^m, ..., \mathcal{K}_k^m] \in \mathbb{R}^{d_{\text{in}} \times k}$ and $\boldsymbol{\mathcal{V}}_m = [\mathcal{V}_1^m, \mathcal{V}_2^m, ..., \mathcal{V}_k^m] \in \mathbb{R}^{|\mathcal{Y}| \times k}$ denote matrices representing all key and value vectors in $\mathcal{N}(\mathbf{h})$, in which we replace the original token value with a one-hot vector for $\mathcal{V}_j^m$. Then, we reformulate the computation of $p_{k\text{NN}}$ in Equation (3):

$$
\begin{aligned}
p_{k\text{NN}}(y_m|\mathbf{x}, \hat{\mathbf{y}}_{<m}) &= \boldsymbol{\mathcal{V}}_m\text{softmax}(\frac{\boldsymbol{\mathcal{K}}_m^\top \mathbf{h}}{T}) \\
&= \text{Attention}(\frac{\boldsymbol{\mathcal{K}}_m}{T}, \boldsymbol{\mathcal{V}}_m, \mathbf{h}),
\end{aligned}
\tag{8}
$$

where we use the dot-product function for the distance metric $d(.,.)$. According to the above equation, $p_{k\text{NN}}$ is a key-value memory with attention storing all nearest neighbors from the datastore.

For the computation of $p_{\text{NMT}}$, we introduce an identity matrix $I_{|\mathcal{Y}|}$ and convert it into attention format:

$$
\begin{aligned}
p_{\text{NMT}}(y_m|\mathbf{x}, \hat{\mathbf{y}}_{<m}) &= \text{softmax}(W_{\mathbf{O}}\mathbf{h}) \\
&= I_{|\mathcal{Y}|}\text{softmax}(W_{\mathbf{O}}\mathbf{h}) \\
&= \text{Attention}(W_{\mathbf{O}}^\top, I_{|\mathcal{Y}|}, \mathbf{h}),
\end{aligned}
\tag{9}
$$

where $W_{\mathbf{O}}^\top = [\text{Emb}_1, \text{Emb}_2, ..., \text{Emb}_{|\mathcal{Y}|}] \in \mathbb{R}^{d_{\text{in}} \times |\mathcal{Y}|}$ is the matrix that represents key vectors for each token in vocabulary. Similarly, $p_{\text{NMT}}$ is a key-value memory with attention storing all representations of the entire vocabulary.

## 3.2 $k$NN-MT as Meta-Optimization

For the ease of qualitative analysis, we follow Irie et al. (2022) to understand the working mechanism of $k$NN-MT under a relaxed linear attention form, i.e., we remove the softmax operation in the computation of $p_{k\text{NN}}$ and $p_{\text{NMT}}$, resulting in the following rewritten expressions for $p_{k\text{NN}}$ and $p_{\text{NMT}}$:

$$
\begin{aligned}
p_{\text{NMT}}(y_m|\mathbf{x}, \hat{\mathbf{y}}_{<m}) &\approx \mathcal{F}_{\text{NMT}}(\mathbf{h}) \\
&= \text{LinearAttn}(W_{\text{O}}^\top, I_{|\mathcal{Y}|}, \mathbf{h}) = W_{\text{O}}\mathbf{h}, \\
p_{k\text{NN}}(y_m|\mathbf{x}, \hat{\mathbf{y}}_{<m}) &\approx \mathcal{F}_{k\text{NN}}(\mathbf{h}) \\
&= \text{LinearAttn}(\frac{\mathcal{K}_m}{T}, \mathcal{V}_m, \mathbf{h}) = \frac{\mathcal{V}_m\mathcal{K}_m^\top \mathbf{h}}{T}.
\end{aligned}
\tag{10}
$$

Then the next-token prediction probability of $k$NN-MT is the weighted sum of two attentions:

$$
\begin{aligned}
p(y_m|\mathbf{x}, \hat{\mathbf{y}}_{<m}) &= \lambda \cdot p_{k\text{NN}} + (1-\lambda) \cdot p_{\text{NMT}} \\
&= p_{\text{NMT}} + \lambda \cdot (p_{k\text{NN}} - p_{\text{NMT}}) \\
&\approx \mathcal{F}_{\text{NMT}}(\mathbf{h}) + \lambda \cdot (\mathcal{F}_{k\text{NN}}(\mathbf{h}) - \mathcal{F}_{\text{NMT}}(\mathbf{h})).
\end{aligned}
\tag{11}
$$

Combing Equation (7), (10) and (11), we derive the dual form between gradient descent-based optimization and $k$NN-MT:

$$
\begin{aligned}
\mathcal{F}_{\text{all}}(\mathbf{h}) =& \mathcal{F}_{\text{NMT}}(\mathbf{h}) + \lambda \cdot (\mathcal{F}_{k\text{NN}}(\mathbf{h}) - \mathcal{F}_{\text{NMT}}(\mathbf{h})) \\
=& W_{\text{O}}\mathbf{h} + \lambda \cdot (\frac{\mathcal{V}_m\mathcal{K}_m^\top \mathbf{h}}{T} - W_{\text{O}}\mathbf{h}) \\
=& W_{\text{O}}\mathbf{h} + \frac{\lambda}{T} \cdot (\mathcal{V}_m\mathcal{K}_m^\top \mathbf{h} - T \cdot W_{\text{O}}\mathbf{h}) \\
=& W_{\text{O}}\mathbf{h} + \frac{\lambda}{T} \cdot (\text{LinearAttn}(\mathcal{K}_m, \mathcal{E}_m, \mathbf{h}) \\
& - \frac{T}{2} \cdot \frac{\partial(\|W_{\text{O}}\|^2)}{\partial W_{\text{O}}}\mathbf{h}) \\
=& W_{\text{O}}\mathbf{h} + \frac{\lambda}{T} \cdot \Delta W_{k\text{NN}}\mathbf{h} \\
=& (W_{\text{O}} + \frac{\lambda}{T} \cdot \Delta W_{k\text{NN}})\mathbf{h} = \mathcal{F}'_{\text{NMT}}(\mathbf{h}),
\end{aligned}
\tag{12}
$$

where $\Delta W_{k\text{NN}} = \mathcal{E}_m\mathcal{K}_m^\top - \frac{T}{2} \cdot \frac{\partial(\|W_{\text{O}}\|^2)}{\partial W_{\text{O}}}$ represents the total gradient including a linear layer (dual form) and $l2$-regularization objective, $\mathcal{K}_m$ stands for nearest-neighbors training inputs to the output projection layer in NMT, and $\mathcal{E}_m = \mathcal{V}_m$ is the corresponding error signals obtained by gradient descent. As shown in the above equations, the introduced probability difference, i.e., $p_{k\text{NN}} - p_{\text{NMT}}$, is equivalent to parameter updates $\Delta W_{k\text{NN}}$ that affect $W_{\text{O}}$. We can also regard $\mathcal{E}_m\mathcal{K}_m^\top = \mathcal{V}_m\mathcal{K}_m^\top$ as some meta-gradients, which are leveraged to compute the updated parameter matrix $\Delta W_{k\text{NN}}$.

In summary, we introduce a new perspective to explain $k$NN-MT as a process of meta-optimization on the output projection layer of NMT, in which $k$NN-MT produces meta-gradients via the computation of $p_{k\text{NN}} - p_{\text{NMT}}$ based on $k$-nearest-neighbors $\mathcal{N}(\mathbf{h}) = \{(\mathcal{K}_j^m, \mathcal{V}_j^m)\}_{j=1}^k$ and implicitly applies gradients to the original output projection layer.

## 3.3 Comparing $k$NN-MT with Fine-tuning

As the Equation (12) indicates that the nearest-neighbors set $\mathcal{N}(\mathbf{h}) = \{(\mathcal{K}_j^m, \mathcal{V}_j^m)\}_{j=1}^k$ serves as the training inputs to the output projection layer in the dual form of $k$NN-MT, we proceed to compare the meta-optimization of $k$NN-MT with explicit OPL fine-tuning. This explicit OPL fine-tuning approach maximizes the log-likelihood of the nearest-neighbors set:

$$
\begin{aligned}
\mathcal{L}(W_{\text{O}}) &= \sum_{j=1}^k \log p_{\text{NMT}}(\mathcal{V}_j^m|\mathcal{K}_j^m) - \frac{\alpha}{2} \cdot \|W_{\text{O}}\|^2 \\
&= \sum_{j=1}^k \mathcal{V}_j^{m\top} \log(\text{softmax}(W_{\text{O}}\mathcal{K}_j^m)) - \frac{\alpha}{2} \cdot \|W_{\text{O}}\|^2,
\end{aligned}
\tag{13}
$$

where $\alpha$ is the hyper-parameter of $l2$-regularization objective and we optimize the parameter matrix of OPL using $\mathcal{K}_j^m$ and $\mathcal{V}_j^m$ as input and label, respectively. By applying the back-propagation algorithm, we obtain the updated matrix $\Delta W_{\text{FT}}$ as follows:

$$
\begin{aligned}
\Delta W_{\text{FT}} &= \frac{\partial \mathcal{L}(W_{\text{O}})}{\partial W_{\text{O}}} \\
&= \sum_{j=1}^k (\mathcal{V}_j^m - \text{softmax}(W_{\text{O}}\mathcal{K}_j^m))\mathcal{K}_j^{m\top} - \alpha \cdot W_{\text{O}} \\
&= \sum_{j=1}^k (\mathcal{V}_j^m - \mathcal{P}_j^m)\mathcal{K}_j^{m\top} - \alpha \cdot W_{\text{O}} \\
&= (\mathcal{V}_m - \mathcal{P}_m)\mathcal{K}_m^\top - \alpha \cdot W_{\text{O}},
\end{aligned}
\tag{14}
$$

where $\mathcal{P}_j^m = \text{softmax}(W_{\text{O}}\mathcal{K}_j^m)$ is the prediction probability of NMT for the context representation $\mathcal{K}_j^m$, $\mathcal{P}_m = [\mathcal{P}_1^m, \mathcal{P}_2^m, ..., \mathcal{P}_k^m] \in \mathbb{R}^{|\mathcal{Y}| \times k}$ represents all prediction probabilities for the entire nearest-neighbours set, and the complete derivation process is presented in Appendix A.1. In the case of standard gradient descent, the new parameter matrix of OPL, i.e., $W'_{\text{O}}$, is computed as:

$$
\begin{aligned}
W'_{\text{O}} &= W_{\text{O}} + \eta \cdot \Delta W_{\text{FT}} \\
&= W_{\text{O}} + \eta \cdot \left((\mathcal{V}_m - \mathcal{P}_m)\mathcal{K}_m^\top - \alpha \cdot W_{\text{O}}\right),
\end{aligned}
\tag{15}
$$

| Methods | Training Data | Error Signals | Gradients | Optimizer |
|---------|---------------|---------------|-----------|-----------|
| $k$NN-MT | $(\mathcal{K}_m, \mathcal{V}_m)$ | $\mathcal{V}_m$ | $\frac{\lambda}{T} \cdot (\mathcal{V}_m \mathcal{K}_m^\top - T \cdot W_O)$ | Computation & Interpolation |
| OPL-FT | $(\mathcal{K}_m, \mathcal{V}_m)$ | $\mathcal{V}_m - \mathcal{P}_m$ | $\eta \cdot ((\mathcal{V}_m - \mathcal{P}_m)\mathcal{K}_m^\top - \alpha \cdot W_O)$ | SGD |

Table 1: The similarities and differences between $k$NN-MT and explicit OPL fine-tuning, where error signals and gradients are provided in Equation (12) and (14).

where $\eta$ is the learning rate. Similar to Equation (12), $\mathcal{K}_m$ denotes training inputs and $\mathcal{E}_m = \mathcal{V}_m - \mathcal{P}_m$ is the corresponding error signals via explicit OPL fine-tuning.

Table 1 displays similarities and differences between $k$NN-MT and explicit OPL fine-tuning, both of which aim to maximize the log-likelihood of a nearest-neighbor set $\mathcal{N}(\mathbf{h}) = \{(\mathcal{K}_j^m, \mathcal{V}_j^m)\}_{j=1}^k$. The main distinction lies in the fact that $k$NN-MT generates meta-gradients through forward computation and interpolation, while fine-tuning computes gradients of OPL through back-propagation. Moreover, we discover that explicit OPL fine-tuning produces gradient formats that are so similar to meta-gradients acquired through $k$NN-MT. Therefore, it is reasonable to view $k$NN-MT as an implicit model fine-tuning process on OPL, in which $k$NN-MT produces a distinct parameter matrix $W_O'$ at each decoding time step. As $k$NN-MT only involves the optimization of OPL compared to entire-model fine-tuning, its performance is evidently constrained by the context representations produced by the base NMT model.

## 4 Experiments

In this section, we begin by comparing the model predictions of $k$NN-MT and explicit OPL fine-tuning (OPL-FT) using multi-domain datasets to verify our earlier analysis. Then we carry out comprehensive multi-domain experiments and word-level analysis to gain a better understanding of the translation performance differences between $k$NN-MT and current popular fine-tuning methods.

### 4.1 $k$NN-MT v.s. Explicit OPL Fine-tuning

**Setup.** We mainly compare $k$NN-MT and OPL-FT on five domain adaptation datasets, including multi-domain German-English datasets in Khandelwal et al. (2021) (IT, Law, Medical, and Koran), and the IWSLT'14 German-English translation dataset. The details of multi-domain datasets are listed in Appendix A.2. The pre-trained NMT model from the WMT'19 German-English news translation task winner (Ng et al., 2019) is used

as the basic model for $k$NN-MT and OPL-FT. We employ both inner-product (IP) and negative $l2$-distance (L2) as distance metrics, in which the datastore size and hyper-parameter settings for $k$NN-MT are included in Appendix A.3 and we maintain consistency with previous work (Zheng et al., 2021a) for most details. As for OPL-FT, the parameter of OPL is trained with the same $k$-nearest-neighbors retrieved by $k$NN-MT via either IP or L2 at each timestep. We perform a grid search and use the perplexity (PPL) on the validation set to determine the optimal learning rate and hyper-parameter for SGD optimization. More details are presented in Appendix A.3. As $k$NN-MT and OPL-FT only involve the optimization of OPL, we adopt a teacher-forcing decoding strategy and evaluate the similarity between them by measuring the mean and variance of the difference between their model predictions on the golden label. Specifically, for the test set containing $n$ target tokens, the mean $M(\cdot)$ and variance $V(\cdot)$ are computed as:

$$M(A - B) = \frac{1}{n} \sum_{i=1}^{n} (p_A(y_i) - p_B(y_i)),$$

$$V(A - B) = \frac{1}{(n-1)} \sum_{i=1}^{n} (p_A(y_i) - p_B(y_i) - M(A - B))^2,$$

where $A, B \in \{\text{NMT}, k\text{NN-MT}, \text{OPL-FT}, \text{FT}\}$ and $p(y_i)$ denotes the model prediction probability on each golden label $y_i$.

**Results.** As shown in Table 2, we find that $k$NN-MT has a more similar model prediction with OPL-FT (lower mean/variance) compared to the base NMT model or entire model fine-tuning (FT). The experimental results indicate that $k$NN-MT and OPL-FT are closer than other tuned models. These findings provide empirical evidence supporting our understanding that $k$**NN-MT performs implicit OPL fine-tuning**. Additionally, we observe that $k$NN-MT achieves a slightly higher mean of model predictions than OPL-FT on average. We suspect that this is because $k$NN-MT solely utilizes the

Table 2 section:

| | | IT | Law | Medical | Koran | IWSLT | Avg. |
|---|---|---|---|---|---|---|---|
| **IP** | $k$NN-MT - NMT | .073 / 0̇37 | .137 / .055 | .133 / .057 | .064 / 0̇41 | **.008** / 0̇14 | .083 / 0̇41 |
| | OPL-FT - NMT | .064 / .043 | .098 / .055 | .100 / .059 | .061 / .044 | .026 / **.011** | .070 / .042 |
| | FT - NMT | .120 / .102 | .147 / .077 | .152 / .093 | .107 / .066 | .038 / .024 | .113 / .072 |
| | FT - $k$NN-MT | 0̇47 / .066 | **.010** / 0̇44 | **.019** / 0̇46 | 0̇43 / .041 | .034 / .044 | 0̇31 / .048 |
| | FT - OPL-FT | .056 / .079 | .049 / .049 | .051 / .056 | .046 / .048 | 0̇12 / .027 | .043 / .052 |
| | $k$NN-MT - OPL-FT | **.010 / .024** | 0̇39 / **.023** | 0̇33 / **.022** | **.003 / .026** | -.018 / **.011** | **.013 / .021** |
| **L2** | $k$NN-MT - NMT | .081 / 0̇37 | .135 / .049 | .137 / .056 | .052 / 0̇37 | .017 / .024 | .082 / 0̇41 |
| | OPL-FT - NMT | .064 / .043 | .098 / .055 | .100 / .059 | .061 / .043 | .026 / **.011** | .070 / .042 |
| | FT - NMT | .702 / .133 | .147 / .077 | .152 / .093 | .107 / .066 | .038 / .024 | .113 / .072 |
| | FT - $k$NN-MT | 0̇39 / .064 | **.012** / 0̇42 | **.016** / 0̇44 | .055 / .040 | 0̇11 / .040 | 0̇27 / .046 |
| | FT - OPL-FT | .056 / .079 | .049 / .049 | .051 / .056 | 0̇46 / .048 | .012 / .027 | .043 / .052 |
| | $k$NN-MT - OPL-FT | **.017 / .024** | 0̇35 / **.018** | 0̇37 / **.022** | **-.019 / .023** | **-.001** / 0̇22 | **.014 / .022** |

Table 2: The mean/variance ($\downarrow$) of the golden label probability differences between base NMT, entire-model fine-tuning (FT), $k$NN-MT and explicit OPL fine-tuning (OPL-FT) over each multi-domain test set.

| Model | # Params | IT | Law | Medical | Koran | IWSLT | Avg. | OOD Avg. | # Speed |
|---|---|---|---|---|---|---|---|---|---|
| NMT | - | 38.35 | 45.48 | 40.06 | 16.26 | 39.12 | 35.85 | 35.36 | 1.00× |
| OPL-FT | 43.03M | 41.26 | 51.51 | 47.56 | 21.27 | 40.50 | 40.42 | 19.62 | 1.00× |
| $k$NN-MT | - | 45.60 | 61.64 | 53.77 | 20.66 | 39.90 | 44.31 | 17.79 | 0.74× |
| AK-MT | 1.2K | 47.40 | 63.32 | 56.38 | 20.77 | 40.04 | 45.58 | 31.69 | 0.72× |
| Adapter($r=64$) | 3.96M | 43.55 | 52.46 | 48.32 | 21.62 | **41.65** | 41.52 | **31.74** | 0.97× |
| Adapter($r=128$) | 7.90M | 44.17 | 53.98 | 49.05 | 21.91 | 41.54 | 42.13 | 31.28 | 0.97× |
| Adapter($r=256$) | 15.77M | 45.27 | 55.55 | 51.32 | 22.38 | 41.57 | 43.22 | 31.06 | 0.95× |
| FT | 269.75M | 49.08 | 63.61 | **58.43** | 22.99 | 41.57 | 47.06 | 22.84 | 1.00× |
| AK-MT$_{\text{Adapter}(r=256)}$ | 15.77M | **49.34** | **64.42** | 57.27 | **23.04** | 41.52 | **47.12** | 29.50 | 0.72× |

Table 3: The BLEU score (%) and decoding speed of all models on multi-domain test sets, including IT, Law, Medical, Koran, and IWSLT. "# Params" refers to the number of fine-tuned parameters. The test sets of the other four domains are integrated as out-of-domain (OOD) test sets for each domain and "OOD Avg." represents the average performance of all models on OOD test sets. For detailed results on the OOD test sets, please refer to Appendix A.4. "# Speed" indicates the relative inference speed using vanilla NMT as a baseline with a batch size of 50k tokens.

label of $k$-nearest-neighbors as error signals to update the models, without considering the prediction of the NMT model, which may weaken the label signal.

## 4.2 Translation Performance

**Setup.** As $k$NN-MT could be viewed as a special case of model fine-tuning, we further compare the translation performance of two $k$NN-based models, i.e., traditional $k$NN-MT and adaptive $k$NN-MT (AK-MT) (Zheng et al., 2021a), with other popular fine-tuning methods, including entire-model fine-tuning (FT) and adapter-based fine-tuning (Adapter). We adopt the previous multi-domain datasets for this experiment but integrate the test sets of the other 4 domains as the out-of-domain (OOD) test set for each domain. The evaluation metric is SacreBLEU, a case-sensitive detokenized BLEU score (Papineni et al., 2002).

All experiments are conducted based on the Fairseq toolkit (Ott et al., 2019). For the Adapter, we build adapter layers according to the approach proposed in Houlsby et al. (2019), with intermediate dimensions $r$ selected from $\{64, 128, 256\}$. For $k$NN-based models, we adopt L2 as the distance metric and the same hyper-parameters as the previous section. We also explore the performance of combining AK-MT and Adapter (AK-MT$_{\text{Adapter}}$), which keeps the same hyper-parameters to AK-MT. The Adam algorithm (Kingma and Ba, 2015) is used for FT, Adapter and OPL-FT[2], with a learning rate of 1e-4 and a batch size of 32k tokens. The training process is executed on 4 NVIDIA Tesla

---

[2]It is difficult to dynamically update the parameter matrix of OPL during beam search, so we directly use all training data to optimize the parameter matrix of OPL in this experiment.

V100 GPUs and the maximum number of training steps is set to 100k with validation occurring every 500 steps. During decoding, the beam size is set to 4 with a length penalty of 0.6.

**Results.**   As illustrated in Table 3, we evaluate the translation performance of all models and obtain the following findings:

- OPL-FT, which optimizes the parameter matrix of OPL, also brings significant improvements. This proves that only updating the parameter of OPL could achieve relatively high domain adaptation performance for NMT since it already produces precise context representation due to the large-scale model pre-training.

- During inference, $k$NN-MT dynamically select the most appropriate training data for optimization at each step, resulting in better performance than OPL-FT. However, $k$NN-MT falls short of FT by 2.75 BLEU score, despite outperforming the Adapter on most domain adaptation tasks. AK-MT achieves better performance than $k$NN-MT but is still weaker than FT, highlighting the necessity of tuning the context representations generated by the original NMT model.

- Combining Adapter and AK-MT achieves comparable translation quality to FT with better performance on OOD test sets (average gain of 6.66 BLEU score). It indicates optimizing the context representations with additional adapter layers could further improve $k$NN-MT.

All in all, as a meta-optimizer on OPL, $k$NN-MT works quite well on domain adaptation tasks but still requires tuning of the context representations generated by the original model to achieve comparable performance to FT.

### 4.3   Word-Level Empirical Analysis

**Setup.**   Apart from the BLEU score, we conduct a word-level analysis to investigate the translation differences between $k$NN-MT and FT, and determine the bottleneck of $k$NN-MT. Specifically, we analyze the translation results of $k$NN-MT, AK-MT, FT, and AK-MT$_{\text{Adapter}}$ by calculating the recall of different target words.[3] We first use spaces as delimiters to extract target words and define the domain-specific degree of each word $w$ as

---

[3]As shown in Appendix A.6, we calculate the precision, recall, and F1 score (**P/R/F1**) for each word in the translation results and observe that the correlation between translation performance and word recall is strongest.

$\gamma(w) = \frac{f_{\text{ID}}(w)}{f_{\text{GD}}(w)}$, where $f_{\text{ID}}(.)$ and $f_{\text{GD}}(.)$ are the word frequencies in domain-specific and general-domain training data, respectively.[4] Then we split the target words into four buckets based on $\gamma$: $\{0 \leq \gamma(w) < 1, 1 \leq \gamma(w) < 2, 2 \leq \gamma(w) < 5, \gamma(w) \geq 5\}$, with words having a higher domain frequency ratio $\gamma$ indicating a higher degree of domain-specificity. To better illustrate the gap between $k$NN-based methods and FT, we define incremental word recall $\Delta\mathbf{R}$ for $k$NN-MT, AK-MT and AK-MT$_{\text{Adapter}}$ as the difference in word recall compared to FT: $\Delta\mathbf{R}(w) = \mathbf{R}(w) - \mathbf{R}_{\text{FT}}(w)$.

**Results.**   Figure 2a presents $\Delta\mathbf{R}$ values for words in different buckets, indicating that compared to FT, $k$NN-MT and AK-MT have poor word recalls for words with $\gamma(w) \geq 2$, particularly when $\gamma(w) \geq 5$. However, AK-MT$_{\text{Adapter}}$ achieves comparable performance to FT, suggesting that enhancing the context representations with adapter layers could handle this issue. Moreover, we focus on words with $\gamma(w) \geq 5$ and evaluate word recalls in different buckets based on word frequency, dividing words into four buckets based on their in-domain frequency ranking: top 1%, top 1~5%, top 5~20%, and top 20~100%. As shown in Figure 2b, **for in-domain low-frequency words, particularly those ranking behind top 20%, $k$NN-MT and AK-MT perform significantly worse than FT in terms of word recall**. Similarly, AK-MT$_{\text{Adapter}}$ yields comparable word recall to FT. These results demonstrate that the performance differences between $k$NN-based models and FT mainly lie in the low recall of in-domain low-frequency words, which can be alleviated by optimizing context representations with additional adapter layers.

**Nearest Neighbors Analysis.**   We verify the performance of $k$NN retrieval for the words with $\gamma(w) \geq 5$ to better understand the quality of context representations. We use the teacher-forcing decoding strategy to calculate the *non-retrieval rate* of words in each bucket, where a word is defined as non-retrieval if any sub-word of it is not retrieved in the $k$-nearest-neighbors of AK-MT and AK-MT$_{\text{Adapter}}$. The $k$-nearest-neighbors of $k$NN-MT and AK-MT are exactly the same. Figure 3 shows that the non-retrieval rate (Unretrieved%) of AK-MT increases as word frequency decreases, consistent with the results of word recall in Figure

---

[4]We manually check the entire dictionary with $\gamma$ and find that most words with $\gamma \geq 2$ are real in-domain words.

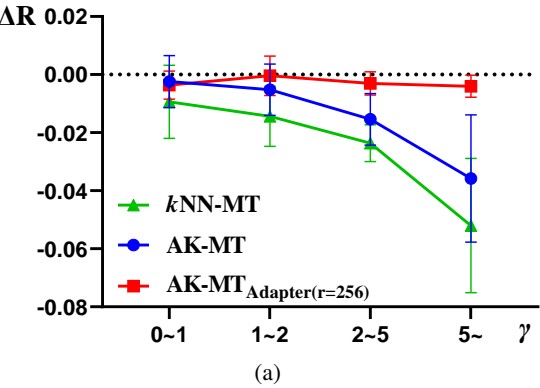 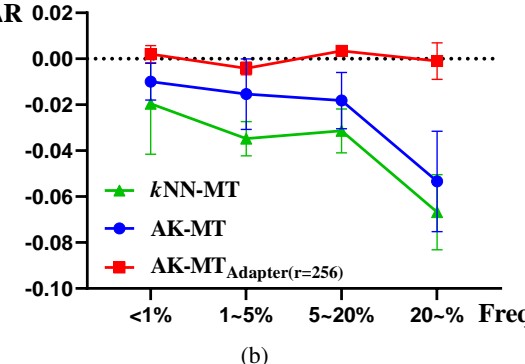

(a)          (b)

Figure 2: Incremental word recall $\Delta\mathbf{R}$ of different words on multi-domain test sets. We plot the mean $\Delta\mathbf{R}$ of five datasets with standard deviation in both figures. For the left figure (a), we count word recalls in different buckets based on $\gamma$, while for the right figure (b), we focus on words with $\gamma(w) \geq 5$ and calculate word recalls in different buckets based on word frequency.

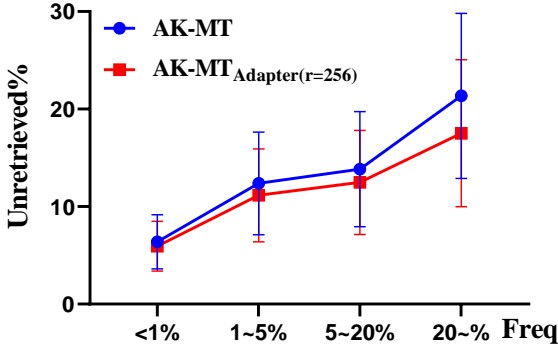

Figure 3: The non-retrieval rate (Unretrieved%) of the words with $\gamma(w) > 5$ on multi-domain test sets. We plot the mean of five datasets with standard deviation.

2b. It indicates that **the context representations of in-domain low-frequency words are not effectively trained at the pre-training stage, resulting in poor word recalls**. With adapter-based fine-tuning, we enhance the context representations for in-domain low-frequency words and thus improve the word recall of AK-MT. We also introduce more metrics to evaluate the property of the nearest-neighbors set and please refer more details in Appendix A.6.

## 5 Related Work

Retrieval-augmented methods have attracted much attention from the community and achieved remarkable performance on various tasks, including language modeling (Khandelwal et al., 2020; He et al., 2021; Nie et al., 2022; Xu et al., 2023; Wang et al., 2023), machine translation (Khandelwal et al., 2021; Zheng et al., 2021a,b; Jiang et al., 2021; Wang et al., 2022b; Du et al., 2022, 2023), question answering (Guu et al., 2020; Lewis et al.,

2020; Xiong et al., 2021), and dialogue generation (Fan et al., 2021; Thulke et al., 2021).

For the NMT system, Khandelwal et al. (2021) propose $k$NN-MT that utilizes a $k$NN classifier over a large datastore with traditional NMT models (Bahdanau et al., 2015; Vaswani et al., 2017; Hassan et al., 2018) to achieve significant improvements. Recently, several attempts have been made by most researchers to improve the robustness and scalability. Meng et al. (2022) and Martins et al. (2022a) propose fast versions of $k$NN-MT. Zheng et al. (2021a) develop adaptive $k$NN-MT by dynamically determining the number of retrieved tokens $k$ and interpolation $\lambda$ at each step, while Martins et al. (2022b) attempt to retrieve chunks of tokens from the datastore instead of a single token. Wang et al. (2022a) adopt a lightweight neural network and the cluster-based pruning method to reduce retrieval redundancy. Dai et al. (2023b) improve both decoding speed and storage overhead by dynamically constructing an extremely small datastore and introducing a distance-aware adapter for inference, and further observe the similar behaviours between kNN-based methods and translation memory approaches (Gu et al., 2018; Zhang et al., 2018; Hao et al., 2023).

Despite the great success of the $k$NN-MT family, the working mechanism of these methods remains an open question. Zhu et al. (2023) analyze the relationship between the datastore and NMT model to better understand the behaviour of $k$NN-MT. To the best of our knowledge, we are the first to provide a meta-optimization perspective for $k$NN-MT, i.e., $k$NN-MT performs implicit gradient descent on the output projection layer.

## 6 Conclusion

In this paper, we present a new meta-optimization perspective to understand $k$NN-MT and establish connections between $k$NN-MT and model fine-tuning. Our results on multi-domain datasets provide strong evidence for the reasonability of this perspective. Additional experiments indicate that (*i*) incorporating $k$NN-MT with adapter-based fine-tuning achieves comparable translation quality to entire-model fine-tuning, with better performance on out-of-domain test sets; (*ii*) $k$NN-based models suffer from the low recall of in-domain low-frequency words, which could be mitigated by optimizing the representation vectors with lightweight adapter layers. We hope our understanding would have more potential to enlighten $k$NN-based applications and model design in the future.

## Acknowldgements

We would like to thank the anonymous reviewers for their insightful comments. Ruize and Rui are with the MT-Lab, Department of Computer Science and Engineering, School of Electronic Information and Electrical Engineering, and also with the MoE Key Lab of Artificial Intelligence, AI Institute, Shanghai Jiao Tong University, Shanghai 200204, China. Rui is supported by the General Program of National Natural Science Foundation of China (62176153), Shanghai Pujiang Program (21PJ1406800), Shanghai Municipal Science and Technology Major Project (2021SHZDZX0102), the Alibaba-AIR Program (22088682), and the Tencent AI Lab Fund RBFR2023012.

## Limitations

In this section, we discuss the limitations and future research directions of our work:

- In the theoretical interpretation of $k$NN-MT, we adopt a relaxed form of attention in the computation of $p_{k\text{NN}}$ and $p_{\text{NMT}}$ for qualitative analysis, following the approach of preview work (Irie et al., 2022; Garg et al., 2022; Dai et al., 2023a). Whether this conclusion is suitable for normal attention is not rigorously proven, but empirical results provide strong evidence of the plausibility of this perspective.

- This paper does not include the results of combining other parameter-efficient fine-tuning methods, such as Prefix-tuning (Li and Liang, 2021) and LoRA (Hu et al., 2022), with $k$NN-MT. But

these methods actually share a similar composition function to optimize the context representations (He et al., 2022). We leave this exploration as the future work.

- The word-level empirical analysis indicates that $k$NN-based models suffer from the low recall of in-domain low-frequency words. Apart from adapter-based fine-tuning, this issue may be mitigated by enhancing the context representations of low-frequency words via more efficient approaches, e.g., introducing frequency-aware token-level contrastive learning method (Zhang et al., 2022) at the pre-training stage and leveraging large-scale pre-trained models (Devlin et al., 2019; Brown et al., 2020; Guo et al., 2020; Li et al., 2022).

- Theoretical and empirical analysis on $k$NN-MT actually could be directly applied to nearest neighbor language models ($k$NN-LM) (Khandelwal et al., 2020). In the future, we would like to follow this research line and do more in-depth explorations on $k$NN-LM. Moreover, the theoretical analysis in this paper is limited to the last hidden states of NMT and we are also interested in investigating the effectiveness of our analysis on other hidden states of NMT, such as the output of the last attention layer in the decoder (Xu et al., 2023).

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

## A  Appendix

### A.1  Derivation Process of $\Delta W_{\text{FT}}$

According to the chain rule, the updated matrix $\Delta W_{\text{FT}}$ is calculated as follows:

$$
\begin{aligned}
\Delta W_{\text{FT}} &= \frac{\partial \mathcal{L}(W_{\text{O}})}{\partial W_{\text{O}}} \\
&= \sum_{j=1}^{k} \frac{\partial (\mathcal{V}_j^{m\top} \log(\text{softmax}(W_{\text{O}} \mathcal{K}_j^m)))}{\partial W_{\text{O}}} \\
&\quad - \frac{\alpha}{2} \cdot \frac{\partial (\|W_{\text{O}}\|^2)}{\partial W_{\text{O}}} \\
&= \sum_{j=1}^{k} \frac{\partial (\mathcal{V}_j^{m\top} \log(\text{softmax}(\mathcal{Z}_j^m)))}{\partial \mathcal{Z}_j^m} \cdot \frac{\partial \mathcal{Z}_j^m}{\partial W_{\text{O}}} \quad (16) \\
&\quad - \alpha \cdot W_{\text{O}} \\
&= \sum_{j=1}^{k} \frac{\partial (\mathcal{V}_j^{m\top} \log(\text{softmax}(\mathcal{Z}_j^m)))}{\partial \mathcal{Z}_j^m} \mathcal{K}_j^{m\top} \\
&\quad - \alpha \cdot W_{\text{O}},
\end{aligned}
$$

where $\mathcal{Z}_j^m = W_{\text{O}} \mathcal{K}_j^m$ and $\frac{\partial \mathcal{Z}_j^m}{\partial W_{\text{O}}} = \mathcal{K}_j^{m\top}$. Then we provide the derivation process for the rest part. Assume that $l$ denotes the vocabulary index of $\mathcal{V}_j^m$, $p_i$ is the $i$-th probability computed by $\text{softmax}(\mathcal{Z}_j^m)$ and $z_i$ stand for the $i$-th value of the vector $\mathcal{Z}_j^m$. The calculation of $\mathcal{F} = \mathcal{V}_j^{m\top} \log(\text{softmax}(\mathcal{Z}_j^m))$ can be re-written as $\mathcal{F} = \log(p_l)$. When $i = l$, the partial derivative of $\mathcal{F}$ to $z_i$ is calculated as:

$$
\begin{aligned}
\frac{\partial \mathcal{F}}{\partial z_i} &= \frac{1}{p_l} \cdot \frac{\partial p_l}{\partial z_i} = \frac{1}{p_l} \cdot \frac{\partial \left( \frac{e^{z_i}}{\sum_{k=1}^{|\mathcal{V}|} e^{z_k}} \right)}{\partial z_i} \\
&= \frac{1}{p_l} \cdot \frac{e^{z_i} \left( \sum_{k=1}^{|\mathcal{V}|} e^{z_k} \right) - (e^{z_i})^2}{\left( \sum_{k=1}^{|\mathcal{V}|} e^{z_k} \right)^2} \quad (17) \\
&= \frac{1}{p_l} \cdot (p_l - p_l^2) = 1 - p_i.
\end{aligned}
$$

If $i \neq l$, we have:

$$
\begin{aligned}
\frac{\partial \mathcal{F}}{\partial z_i} &= \frac{1}{p_l} \cdot \frac{\partial p_l}{\partial z_i} = \frac{1}{p_l} \cdot \frac{\partial \left( \frac{e^{z_l}}{\sum_{k=1}^{|\mathcal{V}|} e^{z_k}} \right)}{\partial z_i} \\
&= \frac{1}{p_l} \cdot - \frac{e^{z_l} \cdot e^{z_i}}{\left( \sum_{k=1}^{|\mathcal{V}|} e^{z_k} \right)^2} \quad (18) \\
&= \frac{1}{p_l} \cdot - p_l \cdot p_i = 0 - p_i.
\end{aligned}
$$

Combining the above equations, we have:

$$
\frac{\partial F}{\partial \mathcal{Z}_j^m} = \mathcal{V}_j^m - \mathcal{P}_j^m, \quad (19)
$$

| | IT | Law | Medical | Koran | IWSLT |
|---|---|---|---|---|---|
| Train | 222,927 | 467,309 | 248,009 | 17,982 | 160,239 |
| Dev | 2,000 | 2,000 | 2,000 | 2,000 | 7,283 |
| Test | 2,000 | 2,000 | 2,000 | 2,000 | 6,750 |

Table 4: Sentence statistics of multi-domain datasets.

| | | IT | Law | Medical | Koran | IWSLT |
|---|---|---|---|---|---|---|
| Datastore Size | | 3.84M | 19.5M | 7.15M | 542K | 3.96M |
| IP | $k$ | 8 | 4 | 4 | 16 | 32 |
| | $\lambda$ | 0.6 | 0.8 | 0.7 | 0.8 | 0.5 |
| | $T$ | 20 | 10 | 20 | 30 | 20 |
| L2 | $k$ | 8 | 4 | 4 | 16 | 32 |
| | $\lambda$ | 0.7 | 0.8 | 0.8 | 0.8 | 0.5 |
| | $T$ | 10 | 10 | 10 | 100 | 50 |

Table 5: The datastore size (number of tokens) and hyper-parameter choices (i.e., $k$, $\lambda$ and $T$) of $k$NN-MT (IP) and $k$NN-MT (L2) in each domain.

where $\mathcal{V}_j^m$ is the one-hot vector whose the $l$-th value is 1, and $\mathcal{P}_j^m = \text{softmax}(W_{\text{O}} \mathcal{K}_j^m)$ is the whole vector of prediction probability. Finally, the Equation 16 is re-written as:

$$
\begin{aligned}
\Delta W_{\text{FT}} &= \sum_{j=1}^{k} \frac{\partial \mathcal{V}_j^{m\top} \log(\text{softmax}(\mathcal{Z}_j^m))}{\partial \mathcal{Z}_j^m} \mathcal{K}_j^{m\top} \\
&\quad - \alpha \cdot W_{\text{O}} \\
&= \sum_{j=1}^{k} (\mathcal{V}_j^m - \mathcal{P}_j^m) \mathcal{K}_j^{m\top} - \alpha \cdot W_{\text{O}} \\
&= (\boldsymbol{\mathcal{V}}_m - \boldsymbol{\mathcal{P}}_m) \mathcal{K}_m^{\top} - \alpha \cdot W_{\text{O}}.
\end{aligned}
$$
$$(20)$$

### A.2  Dataset Statistics

We adopt a multi-domain dataset and consider domains including IT, Medical, Koran and Law, together with IWSLT'14 German-English (DE-EN) dataset in all our experiments. The sentence statistics of datasets are illustrated in Table 4. For the data preprocessing, we use the Moses toolkit to tokenize the sentences and split the words into subword units (Sennrich et al., 2016) using the bpe-codes provided by Ng et al. (2019).

### A.3  Datastore Size and Hyper-parameters

The datastore size of each domain and the choices of hyper-parameters in $k$NN-MT are shown in Table 5, in which we consider grid search on $k \in \{2, 4, 8, 16, 32\}$, $\lambda \in \{0.1, 0.2, \ldots 0.8, 0.9\}$ and $T \in \{5, 10, 20, 50, 100, 150, 200\}$. We maintain the same hyper-parameters for AK-MT but set $k_{\max} = 16$. For OPL-FT, we perform a grid search to find the best learning rate $lr$. The search range

| Dataset | IT | Law | Medical | Koran | IWSLT |
|---------|-----|-----|---------|-------|-------|
| $lr$ | 4e-3 | 6e-3 | 6e-3 | 2e-3 | 1e-3 |

Table 6: The optimal learning rates for explicit OPL fine-tuning based on the perplexity of the validation set.

for all datasets is the same. The search base values are $\{1, 2, 3, 4, 5, 6, 7, 8, 9\}$ and we scale them to 1e-1, 1e-2, 1e-3 and 1e-4 times, i.e., we have $9 \times 4 = 36$ values to search. In Table 6, we present the details of the selected learning rates on five datasets. Once we obtain the optimal learning rate, the hyper-parameter $\alpha \in \{0, 0.01, 0.05, 0.1, 0.5, 1, 5, 10\}$ is further selected by the perplexity on the validation set of each domain.

### A.4 Translation Performance on Out-of-Domain Test Sets

As shown in Table 7, we report the whole out-of-domain results for the experiment in Section 4.2.

### A.5 Translation Performance of Recent Advancements in $k$NN-MT

We provide a comprehensive comparison of translation performance between recent advancements in $k$NN-MT and the methods mentioned in section 4.2. The results are shown in Table 8. The results of FK-MT, EK-MT, CK-MT and SK-MT are excerpted from Dai et al. (2023b).

### A.6 More Details of Word-Level Analysis

We report the overall P/R/F1 results on multi-domain test sets in Table 9. Compared with precision and F1 score, the defect of $k$NN-MT is more obvious on word recall. In addition, as shown in Table 10, we focus on words with $\gamma(w) \geq 5$ and calculate word recalls in different buckets based on word frequency. For the nearest-neighbors analysis, in addition to the non-retrieval rate mentioned in section 4.3, we evaluate the following metrics: ① **Gold Rank/Gold Dist**: the average gold label rank/distance in the top-$k$ list, while taking the rank and distance of the last word in the top-$k$ list (i.e., the farthest neighbor) if unretrieved; ② **#Gold Labels**: the average number of gold labels in the top-$k$ list; ③ **#Labels**: the average distinct labels in the top-$k$ list, indicating the diversity. For in-domain words ($\gamma(w) \geq 5$), the detailed results of $k$-nearest-neighbors analysis in above metrics are shown in Table 11. We observe that after adapter-based fine-tuning, the non-retrieval rate is reduced as the average distance of the gold label increases.

| Model | # Params | IT | Law | Medical | Koran | IWSLT | Avg. |
|---|---|---|---|---|---|---|---|
| NMT | - | 35.10 | 32.54 | 34.74 | 38.55 | 35.87 | 35.36 |
| OPL-FT | 43.03M | 23.56 | 21.01 | 22.81 | 6.97 | 23.73 | 19.62 |
| $k$NN-MT | - | 22.55 | 13.51 | 12.69 | 8.49 | 31.70 | 17.79 |
| AK-MT | - | 32.87 | 27.79 | 29.36 | **33.80** | **34.62** | 31.69 |
| Adapter($r = 64$) | 3.96M | **33.08** | **30.06** | **30.63** | 32.33 | 32.60 | **31.74** |
| Adapter($r = 128$) | 7.90M | 32.87 | 29.51 | 30.23 | 31.58 | 32.20 | 31.28 |
| Adapter($r = 256$) | 15.77M | 32.67 | 28.45 | 30.09 | 31.95 | 32.14 | 31.06 |
| FT | 269.75M | 14.92 | 19.06 | 21.87 | 27.60 | 30.75 | 22.84 |
| AK-MT$_{\text{Adapter}(r=256)}$ | 15.77M | 31.70 | 26.89 | 27.51 | 30.00 | 31.40 | 29.50 |

Table 7: The BLEU score (%) of all models on out-of-domain (OOD) test sets, including IT, Law, Medical, Koran, and IWSLT. "# Params" refers to the number of fine-tuned parameters. The test sets of the other four domains are integrated as OOD test sets for each domain and "Avg." represents the average performance of all models on OOD test sets.

| Model | IT | Law | Medical | Koran | Avg. |
|---|---|---|---|---|---|
| NMT | 38.4 | 45.5 | 40.1 | 16.3 | 35.0 |
| $k$NN-MT (Khandelwal et al., 2021) | 45.6 | 61.6 | 53.8 | 20.7 | 45.4 |
| FK-MT* (Meng et al., 2022) | 45.5 | 56.0 | 53.6 | 21.2 | 44.1 |
| EK-MT* (Martins et al., 2022a) | 44.4 | 57.8 | 51.9 | 20.1 | 43.6 |
| CK-MT* (Martins et al., 2022b) | 44.2 | 59.7 | 53.1 | 19.3 | 44.1 |
| SK-MT* (Dai et al., 2023b) | 46.2 | 62.3 | **57.6** | 19.5 | 46.4 |
| AK-MT (Zheng et al., 2021a) | 47.4 | 63.3 | 56.4 | 20.8 | 47.0 |
| AK-MT$_{\text{Adapter}(r=256)}$ | **49.3** | **64.4** | 57.3 | **23.0** | **48.6** |

Table 8: The BLEU score (%) of recent advancements in $k$NN-MT and the methods mentioned in section 4.2 on multi-domain test sets, including IT, Law, Medical and Koran.

| $\gamma(w)$ | NMT | $k$NN-MT | AK-MT | AK-MT$_{\text{A}}$ | FT |
|---|---|---|---|---|---|
| **IT** | | | | | |
| 0∼1 | 0.597/0.632/0.614 | 0.672/0.656/0.664 | 0.709/**0.681**/0.695 | **0.743**/0.676/**0.708** | 0.730/0.673/0.700 |
| 1∼2 | 0.764/0.746/0.755 | 0.815/0.764/0.789 | 0.815/0.775/0.795 | **0.837/0.778/0.806** | 0.818/0.773/0.795 |
| 2∼5 | 0.680/0.660/0.670 | 0.724/0.701/0.712 | 0.740/0.710/0.725 | **0.761/0.726**/0.744 | 0.757/**0.726**/0.741 |
| 5∼ | 0.634/0.608/0.621 | 0.699/0.681/0.690 | 0.714/0.707/0.711 | **0.735/0.750**/0.742 | 0.746/**0.750/0.748** |
| SUM | 0.676/0.668/0.672 | 0.736/0.702/0.719 | 0.750/0.724/0.737 | **0.774/0.738/0.755** | 0.767/0.735/0.751 |
| **Law** | | | | | |
| 0∼1 | 0.724/0.730/0.727 | 0.784/0.788/0.786 | 0.830/0.808/0.819 | **0.850**/0.810/**0.829** | 0.835/**0.813**/0.824 |
| 1∼2 | 0.820/0.805/0.812 | 0.875/0.862/0.868 | 0.884/0.872/0.878 | **0.891/0.875/0.883** | 0.885/0.869/0.877 |
| 2∼5 | 0.792/0.756/0.774 | 0.851/0.840/0.845 | **0.869**/0.848/0.859 | 0.868/**0.860/0.864** | 0.867/0.858/0.863 |
| 5∼ | 0.787/0.704/0.743 | 0.838/0.814/0.826 | 0.852/0.820/0.836 | 0.854/0.833/0.844 | **0.856/0.835/0.845** |
| SUM | 0.782/0.753/0.767 | 0.839/0.828/0.833 | 0.860/0.840/0.850 | **0.867/0.846/0.857** | 0.862/0.845/0.854 |
| **Medical** | | | | | |
| 0∼1 | 0.640/0.651/0.646 | 0.695/0.716/0.705 | 0.770/0.713/0.740 | **0.797**/0.715/**0.753** | 0.772/**0.725**/0.748 |
| 1∼2 | 0.737/0.729/0.733 | 0.797/0.764/0.780 | 0.822/0.789/0.805 | **0.837/0.795/0.816** | 0.814/**0.795**/0.804 |
| 2∼5 | 0.777/0.731/0.753 | 0.819/0.771/0.794 | 0.848/0.794/0.820 | **0.853/0.796/0.823** | 0.838/**0.801**/0.819 |
| 5∼ | 0.732/0.654/0.691 | 0.792/0.715/0.752 | **0.817**/0.770/0.793 | 0.815/0.781/0.798 | 0.809/**0.790/0.799** |
| SUM | 0.716/0.684/0.699 | 0.774/0.727/0.750 | 0.812/0.763/0.787 | **0.822**/0.769/**0.795** | 0.806/**0.776**/0.790 |
| **Koran** | | | | | |
| 0∼1 | 0.261/0.252/0.256 | 0.677/**0.570/0.619** | 0.645/0.553/0.595 | 0.677/0.556/0.611 | **0.679**/0.562/0.615 |
| 1∼2 | 0.292/0.259/0.275 | 0.680/0.598/0.636 | 0.673/0.597/0.633 | **0.699**/0.605/0.649 | 0.693/**0.616/0.652** |
| 2∼5 | 0.070/0.067/0.068 | 0.562/0.548/0.555 | 0.566/0.547/0.557 | **0.596**/0.569/0.582 | 0.590/**0.577/0.583** |
| 5∼ | 0.082/0.078/0.080 | 0.554/0.521/0.537 | 0.548/0.525/0.536 | **0.582**/0.549/0.565 | 0.575/**0.556/0.566** |
| SUM | 0.175/0.165/0.170 | 0.612/0.557/0.583 | 0.604/0.554/0.578 | **0.635**/0.568/0.600 | 0.630/**0.577/0.602** |
| **IWSLT** | | | | | |
| 0∼1 | 0.687/**0.732**/0.709 | 0.722/0.714/0.718 | 0.710/0.724/0.717 | **0.746**/0.716/**0.731** | 0.742/0.718/0.730 |
| 1∼2 | 0.789/0.786/0.787 | 0.801/0.792/0.796 | 0.800/0.793/0.796 | **0.813**/0.797/**0.805** | 0.809/**0.799**/0.804 |
| 2∼5 | 0.724/0.685/0.704 | 0.728/0.688/0.707 | 0.731/0.690/0.710 | **0.733**/0.700/**0.716** | 0.729/**0.704/0.716** |
| 5∼ | **0.798**/0.591/0.679 | 0.776/0.645/0.704 | 0.788/0.635/0.703 | 0.745/0.703/**0.724** | 0.736/**0.705**/0.720 |
| SUM | 0.744/0.716/0.730 | 0.757/0.722/0.739 | 0.756/0.724/0.739 | **0.765/0.736/0.750** | 0.760/**0.738**/0.749 |

Table 9: Overall P/R/F1 of all models on multi-domain test sets, in which we count P/R/F1 in different buckets based on the domain-specific degree of each word $\gamma(w)$. AK-MT$_{\text{A}}$ is the brief description of AK-MT$_{\text{Adapter}(r=256)}$.

| | # Words | NMT | $k$NN-MT | AK-MT | AK-MT$_A$ | FT |
|---|---|---|---|---|---|---|
| **IT** | | | | | | |
| top 1% | 993 | 0.751 | 0.767 | 0.786 | **0.809** | 0.802 |
| top 1~5% | 1,223 | 0.707 | 0.748 | 0.761 | 0.764 | **0.772** |
| top 5~20% | 1,492 | 0.591 | 0.705 | 0.718 | **0.747** | 0.743 |
| top 20~100% | 2,239 | 0.501 | 0.638 | 0.651 | 0.720 | **0.722** |
| **Law** | | | | | | |
| top 1% | 4,896 | 0.870 | 0.937 | 0.944 | **0.948** | 0.943 |
| top 1~5% | 4,013 | 0.684 | 0.806 | 0.816 | 0.833 | **0.836** |
| top 5~20% | 3,473 | 0.671 | 0.789 | 0.799 | **0.809** | 0.806 |
| top 20~100% | 2,736 | 0.492 | 0.636 | 0.652 | 0.674 | **0.695** |
| **Medical** | | | | | | |
| top 1% | 2,896 | 0.765 | 0.815 | 0.838 | **0.844** | **0.844** |
| top 1~5% | 2,676 | 0.621 | 0.740 | 0.755 | 0.768 | **0.779** |
| top 5~20% | 2,937 | 0.633 | 0.748 | 0.772 | 0.769 | **0.774** |
| top 20~100% | 4,339 | 0.618 | 0.717 | 0.738 | 0.761 | **0.776** |
| **Koran** | | | | | | |
| top 1% | 5,727 | 0.183 | **0.749** | 0.722 | 0.726 | 0.737 |
| top 1~5% | 3,502 | 0.023 | 0.505 | 0.486 | 0.521 | **0.545** |
| top 5~20% | 2,669 | 0.003 | 0.430 | 0.432 | **0.468** | **0.468** |
| top 20~100% | 2,685 | 0.001 | 0.234 | 0.255 | **0.294** | 0.282 |
| **IWSLT** | | | | | | |
| top 1% | 9,774 | 0.669 | 0.751 | 0.718 | 0.790 | **0.791** |
| top 1~5% | 5,856 | 0.519 | 0.599 | 0.566 | 0.634 | **0.640** |
| top 5~20% | 2,243 | 0.540 | 0.565 | 0.563 | **0.608** | 0.603 |
| top 20~100% | 1,497 | 0.522 | 0.547 | 0.552 | 0.623 | **0.631** |

Table 10: The word recall of all models on multi-domain test sets, in which we focus on words with $\gamma(w) \geq 5$ and calculate word recalls in different buckets based on word frequency. "# Words" denotes the total number of examples in different buckets. AK-MT$_A$ is the brief description of AK-MT$_{\text{Adapter}(r=256)}$.

| | Unretrieved% (↓) | | Gold Rank (↓) / Gold Dist (↓) | | #Gold Labels (↑) | | #Labels (↓) | |
|---|---|---|---|---|---|---|---|---|
| | AK-MT | AK-MT$_A$ | AK-MT | AK-MT$_A$ | AK-MT | AK-MT$_A$ | AK-MT | AK-MT$_A$ |
| **IT** | | | | | | | | |
| top 1% | 8.26% | 7.55% | 2.89 / 59.20 | 2.55 / 69.23 | 11.43 | 12.18 | 3.27 | 2.68 |
| top 1~5% | 12.35% | 11.61% | 3.25 / 65.46 | 3.10 / 83.58 | 10.04 | 10.70 | 3.75 | 3.28 |
| top 5~20% | 14.75% | 13.74% | 3.19 / 69.25 | 2.98 / 79.91 | 10.06 | 10.83 | 3.61 | 3.13 |
| top 20~100% | 20.63% | 15.36% | 2.94 / 90.12 | 2.52 / 100.98 | 9.55 | 10.88 | 3.56 | 2.79 |
| **Law** | | | | | | | | |
| top 1% | 1.90% | 1.92% | 1.44 / 29.28 | 1.42 / 23.56 | 14.19 | 14.46 | 1.73 | 1.59 |
| top 1~5% | 5.14% | 5.17% | 2.16 / 50.90 | 2.08 / 51.74 | 11.92 | 12.34 | 2.61 | 2.36 |
| top 5~20% | 6.06% | 5.54% | 2.12 / 59.40 | 2.01 / 61.43 | 11.97 | 12.41 | 2.56 | 2.30 |
| top 20~100% | 10.72% | 8.44% | 1.94 / 89.54 | 1.75 / 90.28 | 12.05 | 12.83 | 2.34 | 1.98 |
| **Medical** | | | | | | | | |
| top 1% | 5.52% | 4.97% | 2.17 / 53.94 | 2.07 / 62.05 | 12.15 | 12.56 | 2.89 | 2.55 |
| top 1~5% | 9.68% | 7.59% | 2.46 / 61.28 | 2.22 / 72.97 | 11.32 | 11.94 | 3.01 | 2.72 |
| top 5~20% | 9.87% | 8.41% | 2.24 / 60.43 | 2.13 / 68.29 | 12.08 | 12.58 | 2.62 | 2.37 |
| top 20~100% | 16.80% | 13.55% | 2.31 / 82.42 | 2.08 / 91.77 | 11.38 | 12.20 | 2.64 | 2.28 |
| **Koran** | | | | | | | | |
| top 1% | 7.81% | 6.95% | 3.00 / 62.56 | 2.68 / 92.52 | 9.85 | 10.79 | 4.07 | 3.21 |
| top 1~5% | 18.10% | 15.13% | 4.48 / 74.33 | 3.99 / 116.09 | 7.39 | 8.26 | 5.01 | 4.21 |
| top 5~20% | 20.57% | 16.90% | 4.31 / 83.25 | 3.80 / 124.37 | 7.45 | 8.35 | 4.86 | 4.15 |
| top 20~100% | 32.85% | 27.15% | 4.33 / 125.08 | 3.87 / 176.16 | 5.63 | 6.50 | 5.38 | 4.66 |
| **IWSLT** | | | | | | | | |
| top 1% | 8.44% | 8.28% | 3.11 / 54.43 | 2.99 / 51.16 | 10.92 | 11.27 | 3.08 | 2.81 |
| top 1~5% | 16.62% | 16.26% | 4.53 / 79.06 | 4.45 / 82.59 | 8.45 | 8.75 | 4.48 | 4.10 |
| top 5~20% | 17.92% | 17.79% | 4.74 / 83.92 | 4.60 / 88.23 | 8.01 | 8.26 | 4.55 | 4.20 |
| top 20~100% | 25.78% | 23.11% | 3.29 / 117.77 | 3.09/129.62 | 8.41 | 8.92 | 4.00 | 3.67 |

Table 11: Detailed results of $k$-nearest-neighbors analysis of in-domain words ($\gamma(w) \geq 5$) on multi-domain test sets. AK-MT$_A$ is the brief description of AK-MT$_{\text{Adapter}(r=256)}$.