# OpenReview forum: "Nearest Neighbor Machine Translation is Meta-Optimizer on Output Projection Layer"
_EMNLP/2023/Conference — EMNLP 2023 Main_

### Official Review · Reviewer_QnkT · 2023-08-02

**Soundness:** 4

**Excitement:**

4: Strong: This paper deepens the understanding of some phenomenon or lowers the barriers to an existing research direction.

**Paper Topic And Main Contributions:**

This paper provides new insights into the working mechanism of kNN-MT as an efficient technique to implicitly execute gradient descent on the output projection layer of NMT, indicating that it is a specific case of model fine-tuning. Then, they conduct multi-domain experiments and word-level analysis to examine the differences in performance between kNN-MT and entire-model fine-tuning.

**Reasons To Accept:**

1. The authors provide new insights into the working mechanism of kNN-MT as an efficient technique to implicitly execute gradient descent on the output projection layer of NMT, indicating that it is a specific case of model fine-tuning. They establish connections between kNN-MT and model fine-tuning. I think it is a interesting find.
2. They give a very detailed mathematical explanation of the methods.
3. They conduct solid experiments and provide detailed analysis.
4. They conduct word-level empirical analysis in Section 4.3, which is a good complement to the main experiment.

**Reasons To Reject:**

I don't see any reason to reject this paper.

**Reproducibility:**

3: Could reproduce the results with some difficulty. The settings of parameters are underspecified or subjectively determined; the training/evaluation data are not widely available.

**Reviewer Confidence:**

3: Pretty sure, but there's a chance I missed something. Although I have a good feel for this area in general, I did not carefully check the paper's details, e.g., the math, experimental design, or novelty.

---

> ### Author Rebuttal · Authors · 2023-08-29
>
> Thank you for taking the time to provide a comprehensive review of our work. We appreciate your thoughtful feedback and constructive comments.

---

### Official Review · Reviewer_4kop · 2023-08-05

**Soundness:** 4

**Excitement:**

3: Ambivalent: It has merits (e.g., it reports state-of-the-art results, the idea is nice), but there are key weaknesses (e.g., it describes incremental work), and it can significantly benefit from another round of revision. However, I won't object to accepting it if my co-reviewers champion it.

**Paper Topic And Main Contributions:**

Nearest Neighbor Machine Translation (NNMT) is one of the new NLP methodologies that has been actively researched in recent years. For sequence generation problems such as language modeling or NMT, each token is usually generated autoregressively during inference. NNMT is a method that creates a datastore with the hidden representation and target token generated with training data, and retrieves the datastore items similar to the current hidden state at each step of inference to supplement the prediction of the NMT model. NNMT has the advantage of improving general translation performance by using a general training datastore, or domain adaptation without fine-tuning by using an in-domain datastore. One of the distinctive variants of NNMT is Adaptive KNN Machine Translation (AK-MT). NNMT is not robust in performance since it is highly dependent on the kNN algorithm hyperparameters. AK-MT shows stable performance by adding a meta-k network to adapt to hyperparameters.

As shown above, NNMT has been widely studied as a useful way to improve inference performance non-parametrically, but there has never been a theoretical discussion of the success of NNMT. In this paper, the authors follow the work of Irie et al. (2022) to show that NNMT is theoretically similar to output projection layer (OPL) fine-tuning, and provide empirical learning results. The authors also provide directions for further improving the NNMT performance through word-level analysis on NNMT's output sentences. (In this paper, they focus on the domain adaptation effect of NNMT).

Following Irie et al. (2022), the authors first characterize that OPL forward propagation (with appropriate relaxation) has a dual form of attention to embedding vectors. The authors also show that the computation of $P_{kNN}$ is equivalent to the computation of the attention to the datastore, so the interpolation of $P_{NMT}$ and $P_{kNN}$ is essentially the computation of the meta-gradient for OPL using the attention to the datastore. To prove this experimentally, they calculate the probability of test set tokens for base NMT, OPL-finetuned model, kNN-MT and show that OPL-FT and kNN-MT produce similar results.

Furthermore, the authors measure the translation performance of kNN-MT and AK-MT with in-domain and out-of-domain test sets. They show that kNN-MT, OPL-FT, and AK-MT obtained better in-domain test BLEUs compared to the adapter models, but their OOD test scores were poor, and their in-domain test BLEUs were also poor compared to FT. These weaknesses are addressed by introducing adapters to AK-MT, which not only improved the in-domain test score but also improved the OOD test BLEU.

As a further analysis of the above learning results, the authors also conduct a word-level analysis of the test set output sequence of each model. The results show that kNN-MT and AK-MT had poorer in-domain vocabulary recall than FT, especially for words with low in-domain frequency. In contrast, AK-MT with adapters achieve almost the same recall as FT. Furthermore, AK-MT show a worse in kNN retrieval rate for in-domain low-frequency words, and the retrieval rate could be improved by introducing an adapter to AK-MT.

**Questions For The Authors:**

Q1
In Table 3, AK-MT and AK-MT with adapter models need to train Meta-k network before starting inference. So is not #Params for those must be stated?

Q2
In Table 2, it seems a bit obvious that probability distance between kNN-MT and OPL-FT are closer than distance between NMT and each of those models, since they uses in-domain training data to fine-tune (or modify) OPL. In my opinion, if the authors want to claim that kNN-MT and OPL-FT are very similar, the authors should provide probability distance between kNN-MT or OPL-FT and another in-domain tuned model (ex. fully Finetuned one) to show kNN-MT and OPL-FT are closer than other tuned models.

Q3.
In Table 2, the authors claim that OPL-FT and kNN-MT similary behave. But in Table 3, the in-domain test scores of those models are quite different (44.31 vs 40.42) Do you have any suggestion for reason to this incongruity?

**Reasons To Accept:**

The paper provided an easy-to-understand summary of previous research, theoretical hypotheses, and various experiments to support them. The Appendix contains additional experimental results to help further understand the analysis. This is also the first theoretical paper to understand how NNMT works, which is currently being studied a lot, and it will serve as a basis for further analysis.

**Reasons To Reject:**

I think some of the claims in the paper need to be supported by more logical evidence. It would be great if the following questions were answered sufficiently. Also AK-MT with adapter achieves best performance among all models in this paper, but it would be worth to note that AK-MT with adapter model has limitations which it has slow inference speed AND must be fine tune adapter beforehand.

**Reproducibility:**

4: Could mostly reproduce the results, but there may be some variation because of sample variance or minor variations in their interpretation of the protocol or method.

**Reviewer Confidence:**

3: Pretty sure, but there's a chance I missed something. Although I have a good feel for this area in general, I did not carefully check the paper's details, e.g., the math, experimental design, or novelty.

---

> ### Author Rebuttal · Authors · 2023-08-29
>
> Thank you for taking the time to provide a comprehensive review of our work. We appreciate your thoughtful feedback and constructive comments. Below, we address the concerns and questions you raised.
>
> > Q1: \#Params for AK-MT.
>
> Thanks for pointing out our omission. The amount of training parameters of AK-MT is 1216, which is much smaller than that of other methods. We will add \#Params for AK-MT and AK-MT with adapter models in our final version.
>
> > Q2: Probability distance between $k$NN-MT or OPL-FT and another in-domain tuned model.
>
> Thank you for providing these valuable comments.
> We have conducted a thorough analysis of the probability distances between $k$NN-MT, OPL-FT, and another in-domain tuned model (i.e., fully fine-tuning, FT).
> As shown in the following two tables, we find that $k$NN-MT and OPL-FT are closer (lower mean/variance) than other tuned models on average.
> These results serve as empirical evidence that supports our understanding: **$k$NN-MT implicitly performs OPL fine-tuning**.
> We will include these results in our final version to further strengthen our argument.
>
> |  Inner Product   | IT                           | Law                                       | Medical                                   | Koran                         | IWSLT                                      | AVG                           |
> |----------------------------------------|----------------------------------------|----------------------------------------------------|----------------------------------------------------|-----------------------------------------|-----------------------------------------------------|----------------------------------------|
> |                      $k$NN-MT - NMT    |$ .073 / \underline{.037}                $|$ .137 / .055                                        $|$ .133 / .057                                        $|$ .064 / \underline{.041}                 $|$ \textcolor{blue}{\textbf{.008}} / \underline{.014}  $|$ .083 / \underline{.041}                $|
> |                      OPL-FT - NMT      |$ .064 / .043                            $|$ .098 / .055                                        $|$ .100 / .059                                        $|$ .061 / .044                             $|$ .026 / \textcolor{blue}{\textbf{.011}}              $|$ .070 / .042                            $|
> |                       FT - NMT          |$ .120 / .102                            $|$ .147 / .077                                        $|$ .152 / .093                                        $|$ .107 / .066                             $|$ .038 / .024                                         $|$ .113 / .072                            $|
> |                       FT - $k$NN-MT     |$ \underline{.047} / .066                $|$ \textcolor{blue}{\textbf{.010}} / \underline{.044} $|$ \textcolor{blue}{\textbf{.019}} / \underline{.046} $|$ \underline{.043} / \underline{.041}     $|$ .034 / .044                                         $|$ \underline{.031} / .048                $|
> |                       FT - OPL-FT       |$ .056 / .079                            $|$ .049 / .049                                        $|$ .051 / .056                                        $|$ .046 / .048                             $|$ \underline{.012} / .027                             $|$ .043 / .052                            $|
> | $k$NN-MT - OPL-FT |$ \textcolor{blue}{\textbf{.010/.024}} $|$ \underline{.039} / \textcolor{blue}{\textbf{.023}} $|$ \underline{.033} / \textcolor{blue}{\textbf{.022}} $|$ \textcolor{blue}{\textbf{.003/.026}}  $|$ -.018 / \textcolor{blue}{\textbf{.011}}             $|$ \textcolor{blue}{\textbf{.013/.021}} $|
>
> |  L2 Similarity  | IT                           | Law                                       | Medical                                   | Koran                         | IWSLT                                      | AVG                           |
> |----------------------------------------|----------------------------------------|----------------------------------------------------|----------------------------------------------------|-----------------------------------------|-----------------------------------------------------|----------------------------------------|
> |                      $k$NN-MT - NMT    |$ .081 / \underline{.037}                $|$ .135 / .049                                        $|$ .137 / .056                                        $|$ .052 / \underline{.037}                 $|$ .017 / .024                                         $|$ .082 / \underline{.041}                $|
> |                      OPL-FT  - NMT     |$ .064 / .043                            $|$ .098 / .055                                        $|$ .100 / .059                                        $|$ .061 / .043                             $|$ .026/\textcolor{blue}{\textbf{ .011}}          $|$ .070 / .042                            $|
> |                       FT - NMT          |$ .702 / .133                            $|$ .147 / .077                                        $|$ .152 / .093                                        $|$ .107 / .066                             $|$ .038 / .024                                         $|$ .113 / .072                            $|
> |                       FT - $k$NN-MT     |$ \underline{.039} / .064                $|$ \textcolor{blue}{\textbf{.012}} / \underline{.042} $|$ \textcolor{blue}{\textbf{.016}} / \underline{.044} $|$ .055 / .040                             $|$ \underline{.011} / .040                             $|$ \underline{.027} / .046                $|
> |                       FT - OPL-FT      |$ .056 / .079                            $|$ .049 / .049                                        $|$ .051 / .056                                        $|$ \underline{.046} / .048                 $|$ .012 / .027                                         $|$ .043 / .052                            $|
> | $k$NN-MT - OPL-FT |$\textcolor{blue}{\textbf{.017/.024}}$|$ \underline{.035} / \textcolor{blue}{\textbf{.018}} $|$ \underline{.037}/ \textcolor{blue}{\textbf{.022}} $|$ \textcolor{blue}{\textbf{-.019/.023}} $|$ \textcolor{blue}{\textbf{-.001}} / \underline{.022} $|$ \textcolor{blue}{\textbf{.014/.022}} $|
>
> > Q3: Why are the in-domain test scores of those models quite different?
>
> This issue is discussed in footnote 1 on page 6 of this paper.
> While evaluating translation performance in Section 4.2, it is difficult to dynamically update the parameter matrix of OPL during beam search like in Section 4.1, so we directly use all training data to optimize the parameter matrix of OPL in this experiment.
>
> Specifically, in Section 3, we conclude that $k$NN-MT and OPL-FT exhibit similar behavior.
> In this context, "OPL-FT" refers to fine-tuning a different output projection layer at each decoding step using the key-value pairs of $k$-nearest neighbors as training data. To ensure consistency in the $k$-nearest neighbors searched by $k$NN-MT and OPL-FT, we employ the teacher-forcing decoding method in the experiments conducted in Section 4.1. However, in Section 4.2, OPL-FT is trained with the entire training set, similar to FT. The reason why $k$NN-MT outperforms OPL-FT in this scenario is that the training data of
> $k$NN-MT dynamically changes at each step.
>
> > Q4: The limitation of AK-MT with adapter.
>
> We will add the limitations of AK-MT with the adapter in Section 7 of this paper.

---

### Official Review · Reviewer_Gf9v · 2023-08-11

**Soundness:** 3

**Excitement:**

3: Ambivalent: It has merits (e.g., it reports state-of-the-art results, the idea is nice), but there are key weaknesses (e.g., it describes incremental work), and it can significantly benefit from another round of revision. However, I won't object to accepting it if my co-reviewers champion it.

**Paper Topic And Main Contributions:**

This paper proposes a novel perspective to understand kNN-MT by describing it as a special case of fine-tuning, specifically a process of meta-optimization on the Output Projection Layer (OPL) of NMT, and establish connections between kNN-MT and model fine-tuning. The
novel perspective on kNN-MT posits that (i) the working mechanism of kNN-MT is to implicitly execute gradient descent on OPL, producing meta-gradients via forward computation based on k-nearest-neighbors, and (ii) explicit fine-tuning on OPL shares a similar gradient format with the meta-gradients obtained by kNN-MT, according to the derivation of back-propagation.

**Questions For The Authors:**

1) A comprehensive comparison is requested between the presented method and the most recent advancements in Fast kNN-MT (https://aclanthology.org/2022.spanlp-1.3.pdf), considering both the translation quality and generation speed as critical parameters for evaluation.
2) An inquiry is raised regarding the effectiveness of the proposed method when applied to kNN-LM (k-Nearest Neighbors Language Models). Should this method prove effective in this context, a thorough discussion on the subject would be deemed valuable and pertinent.

**Reasons To Accept:**

This paper introduces a novel meta-optimization perspective for comprehending kNN-MT (k-Nearest Neighbors Machine Translation), while also forging connections between kNN-MT and general model fine-tuning. Through experimentation on multi-domain datasets, substantial evidence is provided to affirm the validity of this viewpoint. Key findings include: (i) the combination of kNN-MT with adapter-based fine-tuning, which yields translation quality on par with entire-model fine-tuning and exhibits superior performance on out-of-domain test sets; (ii) the identification of a shortcoming in kNN-based models pertaining to low recall of in-domain, low-frequency words, a challenge that can be alleviated through the optimization of representation vectors using lightweight adapter layers.

**Reasons To Reject:**

The authors do not compare their proposed approach with enough recent methods.

**Reproducibility:**

3: Could reproduce the results with some difficulty. The settings of parameters are underspecified or subjectively determined; the training/evaluation data are not widely available.

**Reviewer Confidence:**

4: Quite sure. I tried to check the important points carefully. It's unlikely, though conceivable, that I missed something that should affect my ratings.

---

> ### Author Rebuttal · Authors · 2023-08-29
>
> Thank you for taking the time to provide a comprehensive review of our work. We appreciate your thoughtful feedback and constructive comments. Below, we address the concerns and questions you raised.
>
> > Q1: A comprehensive comparison considering both the translation quality and generation speed as critical parameters for evaluation.
>
> Thank you for pointing out the missing reference [https://aclanthology.org/2022.spanlp-1.3.pdf] and we will ensure to include it in our final version for proper citation.
> While we follow the same experimental setting as the previous paper, we could directly append more comparisons in our final version, including FK-MT, EK-MT, CK-MT and SK-MT (as named in https://openreview.net/pdf?id=uu1GBD9SlLe).
> It is important to note that these methods will be implemented solely on the IWSLT domain. Due to the limited time for response, we are unable to provide the results of these methods on the IWSLT domain.
> We supplement the decoding speed results while batch size is 50k tokens computed based on decoding tokens per second in this Table. We will include it in our final version.
>
> | **Model**                  | **Decoding Speed** |
> |---------------------------------|-------------------------|
> | NMT                             | 1.00x                   |
> | OPL-FT                          | 1.00x                   |
> | $k$NN-MT                        | 0.74x                   |
> | AK-MT                           | 0.72x                   |
> | Adapter($r=64$)                 | 0.97x                   |
> | Adapter($r=128$)                | 0.97x                   |
> | Adapter($r=256$)                | 0.95x                   |
> | FT                              | 1.00x                   |
> | AK-MT$_{\text{Adapter}(r=256)}$ | 0.72x                   |
>
> > Q2: The effectiveness of the proposed method when applied to $k$NN-LM.
>
> As mentioned in our paper, theoretical and empirical analysis conducted on $k$NN-MT actually could be directly applied to $k$NN-LM.
> Specifically, the theoretical reasoning process outlined in Section 3 is fully applicable to $k$NN-LM by replacing $p_{\text{NMT}}$ with $p_{\text{LM}}$.
> Although we currently lack empirical results for $k$NN-LM, we anticipate that it may provide intriguing insights.
> Since this paper primarily focuses on the $k$NN-MT system, we consider the exploration of $k$NN-LM as a potential avenue for future work.

---

### Meta-Review · Area_Chair_uVE8 · 2023-09-19

**Recommendation:** 4

**Metareview:**

This paper investigates the Nearest Neighbor Machine Translation (kNN-MT) method, which has shown success in domain adaptation tasks. The authors aim to provide a theoretical understanding of why kNN-MT works and compare it to entire-model fine-tuning through empirical experiments.

Pros:

* Novel Perspective: The paper introduces a novel perspective by describing kNN-MT as a specific case of model fine-tuning, shedding light on the underlying mechanisms of kNN-MT. This theoretical insight adds depth to the understanding of the method.

* Detailed Analysis: Reviewer 1 appreciates the detailed mathematical explanation of the methods, and Reviewer 3 commends the authors for providing new insights, conducting solid experiments, and offering a comprehensive analysis. The paper's in-depth analysis contributes to its strength.

* Empirical Results: The paper includes extensive empirical experiments, comparing kNN-MT and entire-model fine-tuning. The results are well-presented and provide valuable insights into the performance of these methods in different domains.

* Word-Level Analysis: The paper goes further by conducting word-level analysis, focusing on the recall of in-domain low-frequency words. This adds granularity to the evaluation and highlights the limitations of kNN-MT, which can be addressed with additional adapter layers.


Cons:

* Limited Comparison: Reviewer 1 suggests that the authors could have compared their proposed approach with more recent methods, which could provide a more comprehensive understanding of its strengths and weaknesses.

* Performance Trade-offs: Reviewer 2 notes that while the proposed approach with adapters improves performance, it comes at the cost of slower inference speed and the need for adapter fine-tuning, which are important practical considerations that should be mentioned.


In summary, the paper offers valuable insights into kNN-MT, providing a novel perspective on its working mechanism and conducting thorough empirical and word-level analysis. The paper's strengths lie in its detailed analysis and experimental results. Addressing the suggestions regarding further comparisons and practical considerations could enhance the paper's completeness.

---

### Decision · Program_Chairs · 2023-10-07

**Decision:**

Accept-Main

**Comment:**

This paper investigates the Nearest Neighbor Machine Translation (kNN-MT) method, which has shown success in domain adaptation tasks. The authors aim to provide a theoretical understanding of why kNN-MT works and compare it to entire-model fine-tuning through empirical experiments.

Pros:

* Novel Perspective: The paper introduces a novel perspective by describing kNN-MT as a specific case of model fine-tuning, shedding light on the underlying mechanisms of kNN-MT. This theoretical insight adds depth to the understanding of the method.

* Detailed Analysis: Reviewer 1 appreciates the detailed mathematical explanation of the methods, and Reviewer 3 commends the authors for providing new insights, conducting solid experiments, and offering a comprehensive analysis. The paper's in-depth analysis contributes to its strength.

* Empirical Results: The paper includes extensive empirical experiments, comparing kNN-MT and entire-model fine-tuning. The results are well-presented and provide valuable insights into the performance of these methods in different domains.

* Word-Level Analysis: The paper goes further by conducting word-level analysis, focusing on the recall of in-domain low-frequency words. This adds granularity to the evaluation and highlights the limitations of kNN-MT, which can be addressed with additional adapter layers.


Cons:

* Limited Comparison: Reviewer 1 suggests that the authors could have compared their proposed approach with more recent methods, which could provide a more comprehensive understanding of its strengths and weaknesses.

* Performance Trade-offs: Reviewer 2 notes that while the proposed approach with adapters improves performance, it comes at the cost of slower inference speed and the need for adapter fine-tuning, which are important practical considerations that should be mentioned.


In summary, the paper offers valuable insights into kNN-MT, providing a novel perspective on its working mechanism and conducting thorough empirical and word-level analysis. The paper's strengths lie in its detailed analysis and experimental results. Addressing the suggestions regarding further comparisons and practical considerations could enhance the paper's completeness.